# Household Air Pollution and Angina Pectoris in Low- and Middle-Income Countries: Cross-Sectional Evidence from the World Health Survey 2002–2003

**DOI:** 10.3390/ijerph17165802

**Published:** 2020-08-11

**Authors:** Ishwar Tiwari, Raphael M. Herr, Adrian Loerbroks, Shelby S. Yamamoto

**Affiliations:** 1School of Public Health, University of Alberta, Edmonton, AB T6G 1C9, Canada; shelby.yamamoto@ualberta.ca; 2Mannheim Institute of Public Health, Social and Preventive Medicine, Medical Faculty Mannheim, Heidelberg University, 68167 Mannheim, Germany; Raphael.Herr@medma.uni-heidelberg.de; 3Institute of Occupational, Social and Environmental Medicine, Centre for Health and Society, Faculty of Medicine, University of Düsseldorf, 40225 Düsseldorf, Germany; Adrian.Loerbroks@uni-duesseldorf.de

**Keywords:** angina, angina pectoris, chest-pain, solid fuel, household air pollution

## Abstract

The evidence regarding the effects of household air pollution on angina pectoris is limited in low-and middle-income countries (LMICs). We sought to examine the association between household air pollution and angina pectoris across several countries. We analyzed data of individuals from 46 selected countries participating in the cross-sectional World Health Survey (WHS) 2002–2003. Pooled and stratified (sex, continent) logistic regression with sampling weights was used to estimate adjusted odds ratios (ORs) and 95% confidence intervals (CIs) to quantify associations between the use of different household fuels with angina pectoris. In the pooled sample, we observed lower odds of angina pectoris with electricity use (OR: 0.68, 95% CI: 0.56–0.83) compared to those households reporting the use of gas as a household fuel. Increased odds of angina pectoris were observed with the use of agriculture/dung/shrub/other (OR: 1.65, 95% CI: 1.30–2.09), mixed (solid and non-solid fuels) (OR: 1.31, 95% CI: 1.09–1.56), and mixed solid fuel use (OR: 1.59, 95% CI: 1.12–2.25). Higher odds of angina pectoris were observed mainly with solid fuel use. The results highlight the importance of addressing these issues, especially in regions with a high proportion of solid fuel users and increasing levels of cardiovascular disease.

## 1. Introduction

Approximately 3 billion people worldwide still rely on solid fuels (wood, charcoal, coal, animal dung, grasses, shrubs) and kerosene for cooking and heating purposes [1]. These solid fuels are highly polluting and considered a primary source of household air pollution [2]. Household air pollution persists as a significant public health concern, particularly in low- and middle-income countries (LMICs) [3].

In 2019, the World Health Organization (WHO) reported that air pollution was among the leading threats to global health [4]. Air pollution, including household air pollution, is an established risk factor for several chronic health outcomes such as cardiovascular diseases, including ischemic heart disease (IHD) and stroke, chronic obstructive pulmonary disease, lower respiratory infections like pneumonia, and lung cancer [3,5,6,7]. Household air pollution-related illness is responsible for approximately 3.8 million premature deaths every year; of this, 45% is caused by stroke and IHD [1].

In this study, we focused on angina pectoris as the desired health outcome. Angina pectoris is a type of chest pain caused by reduced blood flow to the heart and is a symptom of coronary artery disease [8]. Globally, the prevalence of age-standardized angina pectoris was 20.3 per 100,000 in males and 15.9 in females in 2010 [9]. The analysis of the data drawn from the WHO’s Study of global AGEing and adult health (WHO SAGE) Wave 1 (2007–2010) cross-sectional study found the overall prevalence of angina pectoris in Russia, India, and South Africa, and China to be 39%, 19%, 9% and 8% respectively. Prevalent stable angina pectoris is a significant contributor to IHD among the non-fatal IHD sequelae: acute myocardial infarction (AMI) and ischemic heart failure. The absolute number of IHD Years of Life Disabled (YLD) increased between 15% to 33% in high-middle-income countries (HMICs), and between 13% to 100% in LMICs from 1990 to 2010 [9]. However, this increase may also be explained by other population changes such as ageing and population growth.

Several studies have explored the impact of ambient air pollution and smoking on angina pectoris [5,6,10,11,12,13,14,15,16,17]. A cohort study in five European cities in Germany, Spain, Finland, Italy, and Sweden observed that ambient air pollution was associated with increased risk of hospital cardiac readmissions of myocardial infarction survivors [5]. Similarly, a retrospective time-series study in Iran also reported positive associations between carbon monoxide (CO) pollutants and hospital admissions due to cardiac angina [18]. Much of this research on air pollution and cardiovascular health has been done in HMICs [5,6,10,11,12,13,16,17]. Less has been done in LMICs [18,19]. Air pollution levels in LMICs are often higher, and the burden of cardiovascular morbidity and mortality is greater than in HMICs, attributable to risk factors associated with the economic transition, population ageing and rapid urbanization [20].

One of the primary sources of exposure for women in LMICs is household air pollution [21,22,23,24,25]. Less is also known about the effects of household air pollution in terms of angina pectoris and whether these effects differ by sex. We aimed to explore sex-stratified relationships between household air pollution and angina pectoris. Using data from the World Health Survey (2002), we investigated the association of household air pollution with angina pectoris in 46 countries.

## 2. Methods

### 2.1. Study Population

We used data from the World Health Survey (WHS), which is a cross-sectional study conducted in 2002–2003 in 70 countries to generate information on the health of adult populations, health care utilization and health systems from over 300,000 individuals [26,27,28,29]. Specifically, the WHS included questions on sociodemographics, economic factors, questions on health status, and questions on whether the respondent had even been diagnosed with five chronic diseases: depression, asthma, arthritis, angina pectoris, and diabetes; whether the respondent had ever received or was currently being treated for these conditions; and a series of symptoms related questions for each condition except diabetes [26,27,28]. The 70 countries represented the six WHO regions: 18 countries in Africa, seven in the Americas, four from the Eastern Mediterranean region, 31 in Europe, and five each from South-East Asia and the Western Pacific regions [29]. The survey population included men and women ≥ 18 years of age who lived in a private-household and were not out of the country during the survey period [30]. A multistage cluster sampling approach was employed in 60 of the 70 countries [28,29,30]. This involved dividing the population into clusters, with one or more clusters randomly clustered and everyone within the chosen cluster sampled. For the rest, study participants were recruited based on single-stage random sampling, which involved the selection of a simple random sample of clusters with data collected from every unit in the sampled clusters. The sampling process was stratified by gender, age, and residence (urban/rural). In 68 countries, data were collected via face-to-face interviews by trained interviewers with at least a high school education [29]. All interviewers underwent a weeklong training course using a standard protocol manual and participated in practice field interviews before data collection [29]. Luxembourg and Israel used telephone surveys to collect data, while face-to-face interviews were carried out in all other countries [29]. The response rates varied between 63% (Israel) and 99% (Philippines). The study was approved by ethics committees at each WHS site, and informed consent was obtained from all study participants [28,30]. For this study, we included responses from 46 countries. Eleven countries were excluded due to single-stage random sampling, while we excluded 13 countries because relevant variables were not collected. The flow chart of the sample selection is presented in the Figure 1.

### 2.2. Exposure

Exposure to household air pollution was assessed by proxy. In the survey, participants were asked about the primary type of fuel used in the household for cooking and heating purposes. Fuel use was examined separately and grouped into the following categories: gas, electricity, and kerosene; coal, charcoal, wood, agriculture, crop, shrub, and grass; and mixed solid, mixed liquid, gas, electricity (mixed fuel use). Fuels were grouped based on evidence from prior studies examining respiratory and cardiovascular symptoms [31,32,33]. Fuel use was self-reported. Data were also collected on the type of stove used (open fire, stove with or without chimney or hood, or closed stove with chimney) for cooking and heating purposes. The WHS survey also asked participants about where they cooked (in the same room used for sleeping or living, in a separate room used as a kitchen, in a separate building used as a kitchen, or outdoors).

### 2.3. Outcome

The outcome examined in this study was angina pectoris, which was determined from a series of questions. These included self-reports of a diagnosis (“Have you ever been diagnosed with angina or AP (a heart disease)?”), self-reported treatment (“Have you ever been treated for it?” or “Have you been taking any medications or other treatment for it during the last two weeks?”), and diagnoses based on the WHO Rose questionnaire [34]. The WHO Rose questionnaire represents a widely accepted approach to angina pectoris assessment in the general population [26]. The WHO Rose questionnaire operationalizes angina pectoris as meeting all criteria such as symptoms worsened by exertion, relieved by rest or nitrates, and located on the sternum, at the left side of the chest or the left arm) during the last 12 months. If participants answered yes to any of these questions, they were coded as having angina pectoris.

### 2.4. Statistical Analysis

We used SPSS version 22.0 (Armonk, NY: IBM Corp, US) to conduct complete case analyses. Post-stratified probability weights were used in the analyses to account for the multistage probability sampling approach. We used logistic regression to estimate odds ratios (ORs) and 95% confidence intervals (CIs) in the stratified (by WHO regions) and pooled samples. We adjusted for several potential confounders; those were age, sex, education (no formal education/primary school completed, secondary school, high school or above), Body Mass Index (BMI) (underweight, normal, overweight, obese), marital status (married/cohabitating, single/separated/divorced/widowed), smoking (never/former, current), alcohol consumption (never/former, current), physical activity (low, moderate, high) and diabetes. Confounders were identified a priori, based on a combination of previous study findings and clinical knowledge [28]. We also carried out analyses stratified by sex. We also ran sensitivity analysis, examining angina pectoris as operationalized by different diagnostic criteria. Specifically, we used different definitions of angina pectoris (i.e., self-report of a diagnosis, self-reported treatment, and WHO Rose questionnaire) and compared the consistency of the results across them.

## 3. Results

Table 1 presents the demographic and clinical characteristics of the study population categorized into four WHO regions (Africa, Americas, Europe, and Asia). The total study population of 229,267 consisted of 47.6% men and 52.4% women with an overall mean age of 40.7 years. Europe and the Americas reported the highest proportion of overweight participants. Overall, more than half of the study population (52.0%) had no formal schooling/completed primary schooling.

Wood is the dominant household fuel used for cooking or heating, used by 37.90% of the study population. Wood use was highest in Asia (55.4%), followed by Africa (49.5%). Wood use was low among households in the Americas (10.3%), and Europe (5.0%), where the principal fuel used was gas (83.6% and 52.3% respectively). Gas as the primary household fuel was infrequently used in Asia (20.5%) and Africa (6.3%). Mixed fuel use was reported by 10.1% of the total study population. Among the regions, the use of mixed fuel was highest in Europe (31.4%), followed by Africa (16.0%), Asia (4.8%) and Americas (3.6%).

Around 8.0% of the study population reported a diagnosis of angina pectoris. Europe (16.8%) reported the highest prevalence and the Americas the lowest (5.2%). About 7.0% of the study population was found to have treated for angina pectoris, with most located in Europe (18.4%). Reports of treated angina pectoris were similar across Africa (4.9%), Americas (4.6%) and Asia (5.4%). In total, 9.00% of the study population was defined as having angina pectoris, according to Rose Angina Questionnaire criteria. The prevalence was highest in Europe (11.3%), followed by Africa (10.4%), Asia (9.5%) and the Americas (4.8%). Overall, the Rose Angina Questionnaire criteria resulted in the highest prevalence of angina pectoris compared to definitions based on those reported as diagnosed or treated.

Approximately 71.0% of the total study population were never/former smokers. Asia region reported the highest percentage of current smokers (34.0%), followed by Europe (28.6%). The highest rate of current alcohol consumption was observed in Europe (73.9%), followed by the Americas (66.7%). High levels of physical activity engagement were reported by 63.8% of the study population. High and moderate levels of physical activity were similar in all the WHO regions, but a greater proportion of the study population in the Americas reported a low level of physical activity (20.4%).

### 3.1. Pooled Results (Angina Pectoris- Diagnosed, Treated, or Rose Criteria)

Overall, fuel use was associated with angina pectoris (diagnosed, treated, or Rose criteria). In the pooled sample, we observed lower odds of angina pectoris with electricity use (OR: 0.68, 95% CI: 0.56–0.83) compared to those households reporting the use of gas as household fuel for cooking or heating. We also observed lower odds with coal/charcoal use (OR 0.80; 95% CI: 0.66–0.97). Increased odds of angina pectoris were observed with the use of agriculture/dung/shrub/other (OR: 1.65, 95% CI: 1.30–2.09), mixed solid and non-solid fuels (OR: 1.31, 95% CI: 1.09–1.56) and solid mixed fuel use (OR: 1.59, 95% CI: 1.12–2.25).

Regionally, electricity (Africa, Europe), kerosene (Europe), and coal/charcoal (Asia) use were associated with lower odds of angina pectoris compared to gas users (diagnosed, treated, or Rose criteria—Table 2). Mixed liquid/gas/electricity use was also associated with lower odds of angina pectoris in Europe (OR: 0.59, 95% CI: 0.41–0.86). Higher odds of angina pectoris were observed with agriculture/dung/shrubs/other fuel use in Africa and Asia. Wood (OR: 1.68, 95% CI: 1.37–2.05) and kerosene (OR: 1.43, 95% CI: 1.05–1.95) use in Africa and Asia, respectively, similarly increased the odds of reported angina pectoris in study participants. Increased odds of angina pectoris were also observed with mixed solid fuel (Africa and the Americas), and mixed liquid/gas electricity use in the Americas compared to gas users.

When associations across the three methods of assessment (diagnosed, treated, Rose criteria) were examined separately, consistent associations were observed with electricity and angina pectoris in pooled analyses (Appendix A). Mixed results were found for other fuels. Across Europe, electricity and mixed non-solid fuel were also consistently associated with lower odds of diagnosed, treated and Rose criteria angina pectoris compared to gas users (Table 2). Wood use was also associated with increased odds of angina pectoris across the three assessment approaches in Africa. Mixed results were observed for other fuel types across the separate regions.

### 3.2. Sex-Stratified Distribution of Angina Pectoris—Diagnosed, Treated or Rose Criteria

Table 3 shows the result of the sex-stratified analyses of the adjusted associations of the self-reported (diagnosed, treated, or Rose criteria) angina pectoris with household fuel type used for heating or cooking in the pooled sample. In men and women, electricity was primarily associated with lower odds of reported angina pectoris compared to those who used gas. Lower odds of angina pectoris were also observed with coal/charcoal use in men and kerosene use in women. Solid and mixed solid fuel use was predominantly associated with increased odds of reported angina pectoris compared to those who used gas for both sexes. Similar results were also observed for mixed (solid/liquid/gas/electricity) household fuel for both men and women.

## 4. Discussions

Electricity use was associated with lower odds of angina pectoris in the pooled sample. Conversely, higher odds of angina pectoris were observed mainly with solid fuel use (agriculture/dung/shrub/other, wood, and mixed solid fuel) in pooled, continental, and sex-stratified analyses. These outcomes were particularly pronounced in Africa, with electricity and non-solid fuels associated with decreased odds of angina pectoris. The strongest associations for household fuel use were observed using the Rose diagnostic criteria. We also observed lower odds of angina pectoris with coal/charcoal use in Asia and pooled analysis.

### 4.1. Household Fuel Use and Angina Pectoris

In the present study, solid fuel use was positively associated with the increased odds of angina pectoris, a finding also observed in prior studies. A cross-sectional study in Nigeria that examined biomass fuel use among women reported that a higher proportion of the women with chest pain (7.5% vs. 1.9%) among those who used biomass fuels compared to those who did not [35]. Adjusted analysis showed that the use of biomass fuels was positively associated with chest pain in women, though the association was not statistically significant (OR: 3.82; *p*-value = 0.09). In contrast, a comparative cross-sectional study in rural villages in Sindh, Pakistan, reported no association between angina pectoris (OR: 1.0, 95% CI 0.8 to 1.4) and the current use of biomass fuel for cooking among women [36]. This study compared women ≥ 40 years who had used biomass fuel for cooking with those who had used non-biomass fuel in the last year, so outcomes from longer-term exposures may have been missed.

### 4.2. Sex Stratified Differences in Angina Pectoris and Household Fuel Use

The present study showed lower odds of angina pectoris with electricity use and higher odds of angina pectoris with solid fuel use in both the sexes. Sex-specific differences were not observed, as was initially hypothesized. Ingale et al. reported that chest pain was significantly associated with agro-waste- and wood-use in female participants [37]. Another study determining the prevalence of angina pectoris and possible angina observed a higher prevalence of angina and possible angina among women 40 years of age or older [38]. Likewise, Hemingway et al. also reported higher excess angina pectoris among women as compared to men in a meta-analysis of studies across 31 countries (pooled random-effects sex prevalence ratio of 1.20, 95% CI 1.14 to 1.28, *p*-value < 0.001) [15].

### 4.3. Ambient Air Pollution and Angina Pectoris

Ambient air pollution and household air pollution share some common air pollutants such as carbon monoxide, nitrogen dioxide, and particulate matter, and cause similar adverse health outcomes at birth, early life, and adulthood [39]. Although studies that have investigated the relationship between household air pollution and angina pectoris were not available, several studies have reported associations between ambient air-pollution and angina pectoris globally [6,12,14,18]. A cross-sectional pilot study in India reported a higher prevalence of the symptoms of angina pectoris and cardiovascular diseases with an increased level of ambient air pollutants [40]. Another study evaluating the influence of ambient air pollution on the number of hospital admissions for acute coronary syndrome in elderly patients reported that a 10 μg/m^3^ increase in PM_10_ was associated with an increase in the number of hospitalizations due to unstable angina [41]. A study in Sao Paulo, Brazil, also observed that an interquartile range increase in CO was associated with a corresponding daily increase in angina or acute myocardial infarction room visits (6.4%, 95% CI: 0.7 to 12.1) [12]. A retrospective time-series study in Tehran also reported a significant relationship between CO level and daily admissions due to angina pectoris [18]. Each unit increase in the CO level was associated with a 1.00,934 increase in the number of hospital admissions (95% CI, 1.00,359–1.01,512) [18]. A prospective cohort study among foundry workers in Finland conducted from 1973 to 1993 by Koskela et al. also observed a dose–response relation in the prevalence of angina pectoris and CO exposure [42].

### 4.4. Smoking and Angina Pectoris

Existing evidence has shown that tobacco smoke is a significant source of indoor air pollution [43]. Therefore, we considered that smoking and indoor air pollution are very similar exposures. The findings from this study are also consistent with evidence from studies on smoking. Secondhand smoke has been linked to several adverse health outcomes, such as acute myocardial infarction and coronary heart disease [10,13,17,44,45]. A study investigating the risk of a first event acute myocardial infarction or unstable angina reported that non-smokers occasionally (<3 times per week) exposed to cigarette smoke had a 26% higher risk (OR: 1.26, *p*-value < 0.01) compared to non-exposed non-smokers [13]. Regular secondhand smoke exposure was associated with a 99% higher risk of developing acute myocardial infarction or unstable angina (OR: 1.99, *p*-value < 0.01) [13]. Lee et al. also reported that passive smoking at home was significantly associated with the risk of angina (RR = 1.016, 95% CI: 1.001–1.032) with the effect more pronounced in urban compared to rural areas [10].

### 4.5. Limitations

Several study limitations must be noted when interpreting these results. The data analyzed in this study are pretty old. Heating and cooking patterns might have changed in LMICs now from what they were before at the time of the original survey. The measurement of angina pectoris in this study relied entirely on self-reports of a key symptom, treatment, and diagnosis. To mitigate this, we performed sensitivity analyses that used different diagnostic criteria in the identification of angina pectoris. Another concern is the assessment of household air pollution exposures. In the present study, the use of the primary type of fuel used for cooking or heating was used as a proxy for household air pollution exposure. The assumption is that certain types of fuels generate different levels of pollutants, with solid biomass fuels generally producing levels higher than that of gas fuels. However, wide variations in exposure can occur across different stove types, stove characteristics, and kitchen locations, which can affect these levels. Mixed fuel and stove use by households is often practiced, which we attempted to capture in this study [46,47,48]. Residual confounding may also be a concern due to the cross-sectional design of the study. Furthermore, the cross-sectional design of the WHS is limited regarding causality inferences.

## 5. Conclusions

This large, multi-country study explored associations between exposure to household air pollution and angina pectoris. Consistent with findings from previous studies, exposure to solid fuels was associated with an increased probability of developing angina pectoris. These results add to the substantial body of the literature regarding the adverse outcomes of air pollution exposure on cardiovascular health. Moreover, the public health significance of these results is that they provide further evidence to support exposure reduction, highlighting the importance of addressing these issues, particularly in those regions with a high proportion of solid fuel users and the increasing prevalence of the cardiovascular disease.

## Figures and Tables

**Figure 1 ijerph-17-05802-f001:**
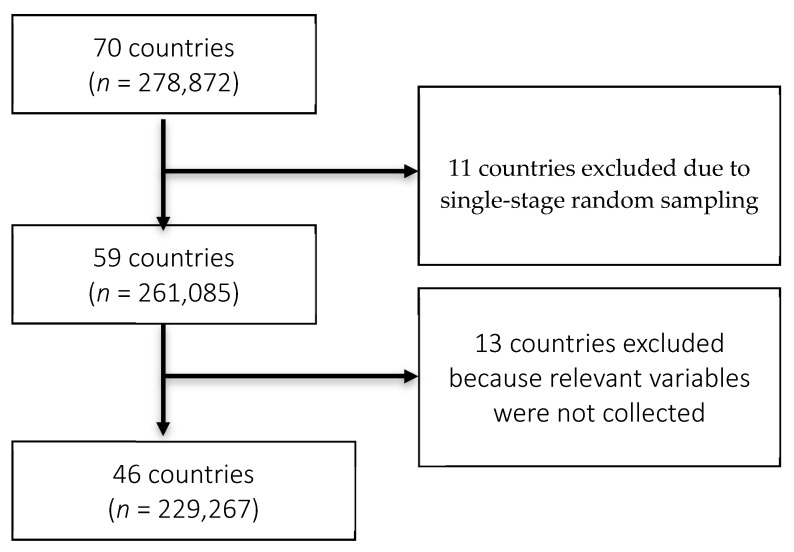
Flow chart.

**Table 1 ijerph-17-05802-t001:** Study population characteristics by continent and pooled total.

Characteristic	Africa (*n* = 69,147)	Americas (*n* = 60,869)	Asia (*n* = 79,567)	Europe (*n* = 19,684)	Total (*n* = 229,267)
Angina pectoris diagnosed (%) ^a^	8.4%	5.2%	6.1%	16.8%	7.9%
Angina pectoris treated (%) ^b^	4.9%	4.6%	5.4%	18.4%	7.3%
Rose angina pectoris (%) ^c^	10.4%	4.8%	9.5%	11.3%	9.0%
Angina pectoris diagnosed, treated or Rose	16.3%	9.5%	14.1%	24.6%	15.2%
Household fuel used for cooking or heating ^d^					
- Gas	6.3%	83.6%	20.5%	52.3%	36.2%
- Electricity	12.8%	0.4%	0.8%	6.3%	2.9%
- Kerosene	3.6%	0.0%	1.7%	3.9%	1.9%
- Coal/charcoal	7.6%	1.8%	6.1%	0.9%	4.6%
- Wood	49.5%	10.3%	55.4%	5.0%	37.9%
- Agriculture/dung/shrubs/other	4.2%	0.2%	10.9%	0.2%	6.4%
- Mixed solid/liquid/gas/electricity	6.9%	1.7%	3.3%	17.5%	5.6%
- Mixed solid only	7.5%	0.1%	0.9%	0.3%	1.4%
Household fuel used for cooking or heating ^e^					
- Gas/electricity/kerosene	22.6%	84.1%	22.9%	62.5%	41.0%
- Coal/charcoal/wood/agriculture/crop/shrub/grass	61.4%	12.3%	72.3%	6.1%	48.9%
- Mixed fuel use	16.0%	3.6%	4.8%	31.4%	10.1%
Mean age (years)	36.84	40.91	38.86	49.00	40.65
Sex (%)					
- Men	49.5%	44.5%	51.1%	38.3%	47.6%
- Women	50.5%	55.5%	48.9%	61.7%	52.4%
BMI (kg/m^2^)					
- Underweight (<18.5)	9.2%	4.2%	14.8%	2.7%	10.1%
- Normal (18.5 ≤ BMI < 25.0)	59.1%	52.7%	57.1%	45.8%	54.4%
- Overweight (25.0 ≤ BMI < 30.0)	20.1%	30.9%	18.4%	35.5%	24.0%
- Obese (BMI ≥ 30.0)	11.6%	12.2%	9.7%	16.0%	11.5%
Marital status (%)					
- Married/cohabitating	58.8%	63.0%	73.7%	57.3%	67.3%
- Single/separated/divorced/widowed	41.2%	37.0%	26.3%	42.7%	32.7%
Highest level of education (%)					
- No formal schooling/primary school completed	69.7%	42.6%	63.5%	14.5%	52.3%
- Secondary school	16.6%	30.1%	15.9%	26.0%	20.3%
- High school or above	13.7%	27.3%	20.6%	59.6%	27.3%
Smoking (%)					
- Never/former	86.3%	77.2%	66.0%	71.4%	71.3%
- Current	13.7%	22.8%	34.0%	28.6%	28.7%
Alcohol consumption (%)					
- Never/former	69.7%	33.3%	83.5%	26.1%	62.8%
- Current	30.3%	66.7%	16.5%	73.9%	37.2%
Physical activity (%)					
- Low	13.6%	20.4%	10.6%	10.3%	12.9%
- Moderate	22.2%	29.0%	20.1%	27.0%	23.3%
- High	64.2%	50.6%	69.3%	62.7%	63.8%
Diabetes (%)	2.5%	5.9%	2.3%	5.0%	3.4%

^a^ Self-report diagnosis of angina pectoris; ^b^ Self-reported treatment of angina pectoris; ^c^ Evaluated according to WHO Rose criteria; ^d^ Mixed use of clean (electricity/gas/kerosene plus coal/charcoal/wood or agriculture/crop/animal dung/shrubs/grass) fuels; ^e^ Mixed use of only solid (coal/charcoal/wood or agriculture/crop/animal dung/shrubs/grass) fuels.

**Table 2 ijerph-17-05802-t002:** Adjusted ^a^ associations (Odds ratios and 95% confidence intervals) of self-reported angina pectoris (Angina diagnosed or treated) or Rose angina pectoris with type of cooking and heating fuels used in household.

Characteristic		Africa		Americas		Asia		Europe		Pooled Sample	
		OR ^a^	(95% CI) ^a^	OR ^a^	(95% CI) ^a^	OR ^a^	(95% CI) ^a^	OR ^a^	(95% CI) ^a^	OR ^a^	(95% CI) ^a^
Household fuel used for cooking or heating	Gas	1		1		1		1		1	
	Electricity	0.72	0.53–0.99	0.61	0.14–2.65	0.74	0.47–1.16	0.44	0.31–0.61	0.68	0.56–0.83
	Kerosene	0.87	0.55–1.37	n/a	n/a	1.43	1.05–1.95	0.21	0.14–0.31	0.84	0.66–1.06
	Coal/charcoal	1.08	0.84–1.39	1.42	0.82–2.46	0.61	0.48–0.79	0.67	0.28–1.59	0.80	0.66–0.97
	Wood	1.68	1.37–2.05	0.86	0.65–1.14	0.91	0.73–1.13	0.10	0.65–1.53	1.14	0.97–1.33
	Agriculture/dung/shrubs/other	1.81	1.06–3.09	1.24	0.57–2.73	1.30	1.01–1.67	1.01	0.33–3.12	1.65	1.30–2.09
	Mixed solid/liquid/gas/electricity	1.55	1.15–2.07	1.46	0.94–2.27	1.08	0.79–1.49	0.91	0.69–1.20	1.31	1.09–1.56
	Mixed solid only	1.99	1.50–2.65	4.08	1.55–10.75	1.14	0.58–2.25	1.06	0.20–5.73	1.59	1.12–2.25
	Mixed liquid/gas/electricity only	1.17	0.73–1.87	2.12	1.13–3.98	0.73	0.50–1.07	0.59	0.41–0.86	0.94	0.69–1.28

Household fuel used for cooking or heating (grouped)	Gas/electricity/kerosene	1		1		1		1		1	
	Coal/charcoal/wood/agriculture/crop/shrub/grass	1.83	1.52–2.21	0.96	0.73–1.25	0.88	0.74–1.05	1.14	0.77–1.70	1.15	1.01–1.31
	Mixed fuel use	1.95	1.57–2.44	1.84	1.28–2.64	1.02	0.78–1.35	0.92	0.73–1.14	1.22	1.06–1.42

OR: odds ratio; 95% CI: 95% confidence interval; ^a^ Adjusted for age, BMI, marital status, highest level of education, smoking, alcohol consumption, physical activity and diabetes.

**Table 3 ijerph-17-05802-t003:** Adjusted ^a^ associations (Odds ratios and 95% confidence intervals) of angina pectoris with type of cook and heating fuels used in household stratified by sex for pooled sample.

Characteristic		Pooled Sample											
		Angina Diagnosed ^b^				Angina Treated ^c^				Rose ^d^			
		Men		Women		Men		Women		Men		Women	
		OR ^a^	(95% CI) ^a^	OR ^a^	(95% CI) ^a^	OR ^a^	(95% CI) ^a^	OR ^a^	(95% CI) ^a^	OR ^a^	(95% CI) ^a^	OR ^a^	(95% CI) ^a^
Household fuel used for cooking or heating	Gas	1		1		1		1		1		1	
	Electricity	0.53	0.38–0.76	0.54	0.39–0.75	0.75	0.52–1.09	0.59	0.43–0.81	0.80	0.53–1.20	0.64	0.45–0.92
	Kerosene	0.85	0.56–1.30	0.62	0.40–0.97	0.80	0.52–1.22	0.57	0.40–0.83	0.93	0.65–1.32	1.13	0.82–1.56
	Coal/charcoal	0.55	0.39–0.80	0.91	0.70–1.19	0.71	0.47–1.05	1.02	0.76–1.37	0.64	0.47–0.87	0.94	0.72–1.21
	Wood	1.06	0.85–1.33	1.16	0.94–1.43	1.02	0.80–1.29	1.03	0.83–1.29	1.35	1.06–1.73	1.16	0.97–1.39
	Agriculture/dung/shrubs/other	1.17	0.72–1.92	0.84	0.49–1.44	1.89	1.10–3.24	0.90	0.55–1.47	2.51	1.69–3.72	1.45	1.02–2.06
	Mixed solid/liquid/gas/electricity	1.10	0.78–1.56	0.95	0.72–1.26	1.31	0.91–1.89	1.17	0.88–1.55	1.70	1.20–2.41	1.32	1.02–1.72
	Mixed solid only	1.21	0.61–2.41	1.20	0.65–2.24	1.26	0.59–2.70	1.12	0.57–2.21	1.42	0.83–2.44	2.35	1.53–3.62
	Mixed liquid/gas/electricity only	1.32	0.62–2.78	0.65	0.47–0.91	1.26	0.61–2.63	0.73	0.53–1.01	1.13	0.61–2.08	0.82	0.62–1.08
Household fuel used for cooking or heating (grouped)	Gas/electricity/kerosene	1		1		1		1		1		1	
	Coal/charcoal/wood/agriculture/crop/shrub/grass	1.04	0.85–1.28	1.15	0.95–1.40	1.07	0.86–1.32	1.08	0.88–1.31	1.35	1.11–1.63	1.15	0.97–1.35
	Mixed fuel use	1.24	0.87–1.78	0.89	0.71–1.12	1.33	0.94–1.88	1.04	0.83–1.30	1.49	1.11–2.01	1.22	1.01–1.47

OR: odds ratio; 95% CI: 95% confidence interval; ^a^ Adjusted for age, BMI, marital status, highest level of education, smoking, alcohol consumption, physical activity and diabetes; ^b^ Self-reported diagnosis of angina pectoris or ^c^ self-reported treatment of angina pectoris or ^d^ Evaluated according to WHO Rose criteria.

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
