# Peer review of "Household Air Pollution and Angina Pectoris in Low- and Middle-Income Countries: Cross-Sectional Evidence from the World Health Survey 2002–2003"

_ijerph, 2020, doi:10.3390/ijerph17165802_

Round 1

Reviewer 1 Report

This study explains the cross-sectional approaches to household air pollution and angina pectoris in low- and middle-income countries. The topic of this paper is significant and appropriate for the IJERPH. In order to make it publishable please consider the following minor suggestions

  1. Despite the fact that the intro gives a good overview about the topic of the paper, it could be improved.

2. Minor spell check required. 

Author Response

Point 1: Despite the fact that the introduction gives a good overview about the topic of the paper, it could be improved.

Response 1: Lines 43-55: Thanks for your comments. We have improved the introduction now.

Point 2: Minor spell check required.

Response 2: Lines 94, 192, 217,224, 225, 238, 267, 291: We have corrected the typo errors.

Reviewer 2 Report

The authors of “Household Air Pollution and Angina Pectoris in low- and middle-income countries: Cross-sectional evidence from the World Health Survey 2002-2003” conducted a cross-sectional study on the association between household air pollution and angina pectoris, and they found that the use of solid fuels was associated with increased risk of angina pectoris. I think this study is well conducted, and I only have minor concerns.

  1. As the authors said, previous studies have suggested the association of ambient air pollution and smoking with angina pectoris. I noticed that they have considered the smoking status as one of the confounders, but why didn’t you adjusted the concentrations of air pollutants such as fine particulate?

  1. I think the total number of participants in the Results section (22,9267) is not agree with the number in Figure 1 (231,005)?

  1. Please add the limitations of cross-sectional study in the Limitation section, and avoid the strong words such as “impact” and “effect” when interpreting the results.

  1. The Conclusion section in this manuscript is just like a shorthand for the whole study, which I think is not right. What does “exposure to solid fuels was associated with increased odds of angina pectoris” exactly means? Moreover, the authors should also state the clinical/public health significance of their results more clearly.

Author Response

Point 1: As the authors said, previous studies have suggested the association of ambient air pollution and smoking with angina pectoris. I noticed that they have considered the smoking status as one of the confounders, but why didn’t you adjusted the concentrations of air pollutants such as fine particulate?

Response 1: Thank you for your comment. Yes, this is true that we considered the smoking status as one of the confounders. As you pointed out, we also recognized that air pollutants such as fine particulate matter as an important source of air pollution in many of these countries, but the reliable ambient PM2.5 data was not available. Therefore, we could not adjust it in our analysis.

Point 2: I think the total number of participants in the results section (229,267) is not agree with the number in figure 1 (231,005)

Response 2: Thank you. It was a mistake. The total number of participants is now corrected in figure 1 (Line 100).

Point 3: Please add the limitations of cross-sectional study in the limitation section, and avoid the strong words such as “impact” and “effect” when interpreting the results.

Response 3: The limitations of a cross-sectional study has been added to the limitation section (Lines 285-286).

The strong words such as “impact” and “effect” have been replaced with “outcome (Lines 216, 230, 244, 292).”

Point 4: The conclusion section in this manuscript is just like a shorthand for the whole study, which I think is not right. What does “exposure to solid fuels was associated with increased odds of angina pectoris” exactly means? Moreover, the authors should also state the clinical/public health significance of their results more clearly.

Response 4: The sentence “exposure to solid fuels was associated with increased odds of angina pectoris” has been reworded in the conclusion section (Lines 290-291).

The public health significance of the study results is now clearly stated (Lines 293-297).

Reviewer 3 Report

Dear authors,

your article is very interesting and I believe that it deserves to be published on the IJERPH.

I just have some suggestion which could improve the understandability for the potential readers.

lines 48-50: please, clarify the meaning of IHD

line 99: please, could you give more details on the exposure assessment? In fact, it is a crucial point of your article and I believe it deserves more explanations.

lines 126-128: please, could you provide more details on the method used to perform the sensitivity analysis?

Table 1: you should reduce the dimension of this table. Maximum 1 page. Take a look at other articles.

The remaining parts of the manuscript are clearly written and I have no comments for you about them.

Regards and good luck!

Author Response

Point 1: lines 48-50: please, clarify the meaning of IHD

Response 1: IHD stands for Ischemic Heart Disease. It is first used in lines 37-38.

Point 2: line 99: please, could you give more details on the exposure assessment? In fact, it is a crucial point of your article and I believe it deserves more explanations.

Response 2: Further details on the exposure assessment have been added to the manuscript (Lines 107-112).

Point 3: lines 126-128: please, could you provide more details on the method used to perform the sensitivity analysis?

Response 3: Further details on the method used to perform the sensitivity analysis have been added to the manuscript (Lines 136-138).

Point 4: Table 1: you should reduce the dimension of this table. Maximum 1 page. Take a look at other articles.

Response 4: The size of table 1 is reduced to one page.

Reviewer 4 Report

while the paper has set to highlight on the use of varied energy type and the risk of developing angina in LMIC however considering the age (18years) of the data, the  scientific soundness of the result have been preceded by time. 

It will have been much better if more recent data is used to compare the trend with what is presented herein. Without doing so, it will be difficult to see rationale and possible impact around the paper.

It is not enough making limitation statement around this but in my opinion, up to date data will be required to compare and possibly show any trend in increase/decline as you have earlier stated in the introduction part. 

In addition, there are few typo errors dotted round the paper which can be improved with further proof read.  

Author Response

Point 1: It will have been much better if more recent data is used to compare the trend with what is presented herein. Without doing so, it will be difficult to see rationale and possible impact around the paper.

Response 1: Thank you for your comments. We agree but that we wanted to examine across several regions that were not available in the more recent data. We also wanted to ensure that we had sufficient power to detect effects if they existed. Also, this study is exploratory and meant to generate hypotheses for future work.

Point 2: It is not enough making limitation statement around this but in my opinion, up to date data will be required to compare and possibly show any trend in increase/decline as you have earlier stated in the introduction part.

Response 2: This is true. We might consider doing another follow-up paper that uses newer data too in the future.

Point 3: In addition, there are few typo errors dotted round the paper which can be improved with further proof read. 

Response 3: We have corrected the typo errors (Lines 94, 192, 217,224, 225, 238, 267, 291).

Round 2

Reviewer 4 Report

I still have my reservation to the justification offered around the use of old data to inform the paper development. However i look forward reading more around  new paper that considered the updated data.